# Evaluating the Feasibility of a Low-Field Nuclear Magnetic Resonance (NMR) Sensor for Manure Nutrient Prediction

**DOI:** 10.3390/s22072438

**Published:** 2022-03-22

**Authors:** Xiaoyu Feng, Rebecca A. Larson, Matthew F. Digman

**Affiliations:** Department of Biological Systems Engineering, University of Wisconsin-Madison, 460 Henry Mall, Madison, WI 53706, USA; rebecca.larson@wisc.edu (R.A.L.); digman@wisc.edu (M.F.D.)

**Keywords:** dairy manure, nutrient, low-field NMR, sensor, nitrogen, phosphorus

## Abstract

Livestock manure is typically applied to fertilize crops, however the accurate determination of manure nutrient composition through a reliable method is important to optimize manure application rates that maximize crop yields and prevent environmental contamination. Existing laboratory methods can be time consuming, expensive, and generally the results are not provided prior to manure application. In this study, the evaluation of a low-field nuclear magnetic resonance (NMR) sensor designated for manure nutrient prediction was assessed. Twenty dairy manure samples were analyzed for total solid (TS), total nitrogen (TN), ammoniacal nitrogen (NH_4_-N), and total phosphorus (TP) in a certified laboratory and in parallel using the NMR analyzer. The linear regression of NMR prediction versus lab measurements for TS had an R^2^ value of 0.86 for samples with TS < 8%, and values of 0.94 and 0.98 for TN and NH_4_-N, respectively, indicating good correlations between NMR prediction and lab measurements. The TP prediction of NMR for all samples agreed with the lab analysis with R^2^ greater than 0.87. The intra- and inter-sample variations of TP measured by NMR were significantly larger than other parameters suggesting less robustness in TP prediction. The results of this study indicate low-field NMR is a rapid method that has a potential to be utilized as an alternative to laboratory analysis of manure nutrients, however, further investigation is needed before wide application for on farm analysis.

## 1. Introduction

Livestock manure contains a variety of essential nutrients including nitrogen (N), phosphorus (P), potassium (K), and micronutrients that are valuable when land applied in maintaining soil fertility and increasing crop production. Although applying manure as a fertilizer at agronomic rates has a great benefit to agricultural systems, over-application of manure may cause nutrient saturation in soils and result in adverse impacts on the environment such as eutrophication and groundwater contamination [1,2]. To minimize the risk of over-application and to ensure that there are sufficient nutrients for optimum crop yields, obtaining the accurate composition of manure through a reliable method is important for crop producers.

Manure is a non-homogeneous mixture with large variations in composition. Analysis of manure characteristics is traditionally completed by wet chemical analysis. Manure analysis conducted by a certified laboratory is regarded as having a high accuracy and is widely applied in agricultural research and production systems [3]. However, laboratory analysis is time-consuming and cost-intensive which is impractical for on-farm testing. Innovative technology that enables a rapid, low-cost, and accurate analysis of manure constituents at the time of application has potential to improve manure nutrient application [4]. Rapid methods have been proposed to analyze physical and chemical characteristics and determine nutrient content of animal manure. Conductimetry is one of the methods that has been investigated in recent studies [5,6,7]. This approach is based on electrical conductivity (EC) measurements which have been observed to be correlated to nutrient contents present in manure slurry. While an EC method is able to give reliable estimates of nutrient contents in animal slurries, it is generally only accurate in determining soluble constituents. An alternative approach that has been recently commercialized for manure nutrient analyses is near-infrared spectroscopy (NIRS) [3,8,9,10]. However, NIRS is an indirect measurement of nutrient content and requires significant calibrations to correlate manure composition with spectral data. These calibrations need maintenance to obtain accurate and precise analyses, which is expensive and technically difficult [3].

A new device based on a low-field nuclear magnetic resonance (NMR) has been introduced in Europe which offers direct measurement of manure compositions including total solid (TS), pH, total nitrogen (TN), ammoniacal nitrogen (NH_4_-N), total phosphorus (TP), and total potassium (TK). This sensor has been investigated in previous research as an alternative to the nutrient monitoring techniques of EC and NIRS [11]. The NMR device is designed based on the absorption and emissions of energy in the range of radio frequency of the electromagnetic spectrum [12]. The resonance frequencies are dependent on the magnetic properties of the relevant nuclei, its surroundings, and the magnetic field strength of the magnet employed. The NMR unit sends strong but very short radio frequency pulses matching the resonance frequency of the nuclei to be measured into the manure sample and receives the response from the nuclei after or in between pulses [13].

Several NMR sensor-based techniques have been applied in different agricultural fields in recent years. NMRs have been widely used in the milk industry to monitor milk processing [12] and a recent application of time domain low-field NMR during milk coagulation, cutting, and syneresis has been reported [14]. Nikolskaya et al. applied a time-domain NMR device to determine the solid content of black liquor and concluded that the inline NMR system had the potential to be applied as an industrial sensor for providing rapid and robust measurements of solid contents [15]. A portable low-field NMR sensor developed for detection of catalytic fines in fuel oil has been demonstrated as a robust and low-cost alternative to laboratory measurements in the shipping industry [16].

The performance of an NMR sensor to determine nutrient content was also evaluated for use in animal slurries in Denmark. The NMR sensor consisted of a 1.5 T Halbach magnet, a digital FPGA (field-programmable gate array) console, a power amplifier, and a shielded probe to obtain frequencies in the MHz range. The relevant isotopes of ^14^N, ^31^P, ^39^K, and ^17^O detection were chosen to determine the NH_4_-N, P, K, and organic N (and dry matter), respectively. NMR signals were acquired using ^14^N Quadrupolar Carr–Purcell–Meiboom–Gill (QCPMG), ^31^P Carr–Purcell–Meiboom–Gill (CPMG), and ^1^H CPMG experiments [11]. Results demonstrated this multinuclear NMR sensor was in good agreement with laboratory analysis for predicting NH_4_-N, organic N, TP, and K contents [11]. The NMR sensor offers major advantages of using non-sensitive parts contact with manure samples and operating without calibration requirements [11].

Although the study conducted by Sørensen et al. [11] has shown potential for NMR application for manure analysis, this new technique for manure sensing has not been fully assessed. This single experiment was conducted with limited sample size and a restricted region. The accuracy and precision of this new NMR technique requires further evaluation before use for the purpose of manure application. NMR analysis is an analytical method of low sensitivity, however, the NMR signal scales linearly on repeating the experiments to collect extra data, whereas the noise scales only with the square root of the number of experiments. Therefore, the sensitivity or precision of the analysis can theoretically be improved with longer run times. The objective of this study was to conduct an experiment to assess the accuracy and precision of a low-field NMR manure sensing device for predicting TS, NH_4_-N, TN, and TP with different run times in dairy manure and to demonstrate the reliability of using NMR for on-farm monitoring and as an alternative to laboratory analysis.

## 2. Materials and Methods

### 2.1. Manure Sample Collection

A total of twenty manure samples were collected from the five dairy farms in Wisconsin between October 2020 and March 2021, labeled S1 to S20. A previous study conducted in Denmark used 16 animal slurries to analyze the NMR sensor for online monitoring of NPK in manure [11]. Approximately 2 L of each sample was collected, and all samples were stored in closed containers at 4 °C for less than 7 days until subsampling and analyses were performed. Previous research has demonstrated that manure samples can be kept for several months at this temperature without impacting the relevant components [17]. During subsampling, the manure was mixed on a stirring plate for 1 h and a 500 mL and 50 mL subsamples were prepared for certified laboratory analysis and NMR measurement, respectively.

### 2.2. Laboratory and NMR Analysis

Chemical analyses for TS, TN, NH_4_-N, TP, TK, and ash of the manure samples were performed at the University of Wisconsin Soil and Forage Laboratory in Marshfield that is certified by the Minnesota Department of Agriculture in the Manure Analysis Proficiency (MAP) Program [18]. The MAP certified laboratory analyzes manure samples following a robust statistical analysis and scoring system and the certification requires to be renewed yearly according to the accuracy and precision of the laboratory which has been proved to be accurate, precise, and accountable [18]. The median absolute deviation (MAD) of manure analysis based on eight MAP certified laboratories of TS, NH_4_-N, TN, P_2_O_5_, and K_2_O were 0.15%, 0.09 g kg^−1^, 0.12 g kg^−1^, 0.38 g kg^−1^, 0.07 g kg^-^1, respectively [2]. Samples were analyzed following the A3769 standard [19].

All NMR measurements were made with a Tveskaeg^TM^ Benchtop NMR analyzer (NanoNord A/S, Aalborg, Denmark). The system contained a shielded strong magnet with a homogeneous magnetic field and a radio-frequency antenna, digital signal processing units, and a computer platform to carry out NMR analysis. All parameters measured, including TS, pH, NH_4_-N, TN, and TP, had been pre-calibrated by the manufacturer by measuring a reference sample with a known concentration. The system was warmed up before use to the normal operating temperature range of 18–28 °C. The inside measurement chamber was kept at 39 °C. The range of transmitting frequencies of the NMR signal is approximately 2–70 MHz. Measuring accuracy and precision on the reference sample, depending on parameter and measuring time of the factory calibrated NMR analyzer, is ±3% and 0.1 mg/L to 10 g/L, respectively [13]. Before introducing samples into the instrument, 50 mL subsamples were blended with an OMNI GLH homogenizer at 3000 rpm for 1 min to break large particles to less than 0.5 mm. Homogenized manure was then transferred into a specifically designed test tube consisting of a perfluoroalkoxy (PFA) tubing with defined dimensions and a set of polytetrafluoroethylene (PTFE) plugs and inserted into the measurement chamber for analysis. The NMR analyzer had a default run time (RT) setting of 10 s for TS and pH. TN and NH_4_-N were analyzed at RTs of 15 min, 30 min, 45 min, and 60 min, and TP at RTs of 30 min, 45 min, 60 min, and 90 min, respectively. The precision or standard deviation (SD) of licensed parameters provided by the manufacturer is listed in Appendix A. Five measurements were taken for each sample of different parameters and RTs.

### 2.3. Data Analysis

Laboratory results reported in lbs/1000 gal were first converted to mg L^−1^ and P as phosphate (P_2_O_5_) and K as potash (K_2_O) were calculated to be represented as TP (mg L^−1^) and TK (mg L^−1^). Data collected from the NMR analyzer were indicated in mg L^−1^ except for TS and pH. The absolute differences (AbsDiff = ǀLab − NMRǀ) between laboratorial results and NMR measurements were calculated for all data points and used for data analysis in this study. Sample mean, standard deviation (SD), minimum, maximum, and coefficient of variation (CV) were based on 5 replicates (20 replicates for TS) of each sample and calculated for both NMR data and AbsDiff data. Overall mean, standard deviation, minimum, maximum, and CVs of different parameters (TS, NH_4_-N, TN, and TP) and RT (15, 30, 45, 60, and 90 min) were based on AbsDiff of 20 samples. These data were analyzed to evaluate the accuracy and precision for test groups. The intra- and inter-sample variations were also reported for analysis of precision. The mean CV from each parameter and RT group was used to express the overall mean intra-sample CV and designated as the repeatability (R_p_). The inter-sample CV which was calculated as the CV among the mean values from all samples within the specific parameter and RT group was used to represent the inter-sample precision and designated as the reproducibility (R_d_). All the R_p_ and R_d_ statistics were analyzed using AbsDiff. Linear regression models for each parameter measured were developed to analyze the relationship between NMR prediction and laboratory measurements. Residuals were calculated as the differences between the NMR measurements and linear model predictions. Samples were further grouped into two different TS levels to evaluate the improvement of the linear correlation between NMR prediction and lab measurements for manure parameters based on the residual analysis.

A sample was considered an outlier when the sample CV exceeded three times its corresponding group CV. Sample means outside the confidence interval of the group mean, ±3 times the group SD, were excluded for poor accuracy [20]. One sample (S11) was flagged for TS; S11 was deducted for TN at all run times; S13 was removed for NH_4_-N at 60 min run time; and S20 was excluded for TP at 30 min and 90 min run times. Significance analyses were performed using one-way analysis of variance (ANOVA) *t*-test (*p* = 0.05). Data analysis was conducted using Microsoft Excel (2014) and RStudio Version 1.2.1335.

## 3. Results

### 3.1. Manure Characteristics

The constitutions of manure nutrients are summarized in Table 1. The solid contents of the samples varied highly with a range between 1.4% and 19.8% (mean ± SD: 6.80 ± 4.74%) due to the differences of animal feed, housing, timing, sampling places, and manure handling systems. The ranges of TN and NH_4_-N were 1114–5824 mg L^−1^ (mean ± SD: 2814 ± 1253.6 mg L^−1^) and 751–4253 mg L^−1^ (mean ± SD: 1301 ± 747.6 mg L^−1^), and S15 and S12 had the lowest and highest concentrations of NH_4_-N and TN, respectively. The concentration of TP varied from 115 mg L^−1^ of S16 to 1097 mg L^−1^ of S20 (mean ± SD: 506 ± 239.1 mg L^−1^). The TS, TN, and TP of dairy manure samples analyzed in this study roughly represented the contents of liquid dairy manure reported in Wisconsin which were 6.0% for TS, 2876 mg L^−1^ for TN, and 471 mg L^−1^ for TP [21]. This indicated that the manure samples selected in this study were representative of the liquid dairy manure characteristics in Wisconsin and rational for analyzing purpose.

### 3.2. Comparison between NMR Prediction and Lab Measurement

#### 3.2.1. Total Solids

Sample 11 was flagged based on the designated outlier criteria which had the highest TS of 19.8% among all samples. Figure 1a provides the linear regression of TS measurements between NMR prediction and laboratory analysis. The R^2^ was 0.80 with the slope and intercept of 1.59 and 0.02, respectively (Table 2), which indicated the overestimation of TS predicted by NMR compared to laboratory estimates. The regression fit (Figure 1a) and residuals between NMR predicted and lab measured of TS (Appendix A) showed as the TS increased to over 8%, the linearity of regression decreased which indicated less accuracy of NMR prediction for manure samples with relative high TS. This may provide evidence that the NMR measurement for TS was affected by manure characteristics and the accuracy and precision of the NMR analyzer can be impaired at higher TS.

To examine the effect of TS levels, the linear regression between NMR prediction and lab measurement was modified based on two TS groups (Figure 1b). As shown in Table 2, R^2^ for TS < 8% was improved to 0.86 and the slope and intercept were 1.34 and 0.04, respectively. For samples with TS > 8%, the linear fit between NMR prediction and lab measurement of TS was significantly worse compared to TS < 8% with the R^2^ = 0.50. The results indicated that the accuracy and precision of the NMR analyzer for predicting TS is affected by the solid content range of manure samples and was more accurate for samples with relatively low TS. The NMR manufacturer assumed ^17^O intensity is correlated to the moisture content and thus total solid content can be determined. However, the complexity of manure composition may introduce bias to the relationship between ^17^O intensity and dry matter. Moreover, the factory calibration of TS may not well represent the range of dry matter content of dairy manure in United States.

#### 3.2.2. Total Nitrogen

Total nitrogen was analyzed using NMR at RT 15, 30, 45, and 60 min, and results in Figure 2 and Table 2 show the linear regression analysis of NMR prediction for TN. The R^2^ values of NMR prediction versus lab measurement for TN at all RTs were less than 0.7 which indicated the predicted vs. measured TN was not in good agreement (Table 2). As RT increased from 15 to 45 min, the R^2^ of linear fit was slightly improved from 0.56 to 0.66 but dropped to 0.61 when RT was continuously increasing to 60 min in Table 2. The slopes of the regression indicated the NMR predictions for TN were overestimated compared to the lab analysis.

Several large residuals were observed for samples with TS greater than 6% (Appendix A). As the NMR performance of TS prediction was highly affected by the TS levels, the effect of high and low TS levels on other parameters was evaluated to examine this assumption. Figure 2b shows the linear fit of TN measurements between NMR analyzer and lab analysis for the two TS levels (TS < 8% and TS > 8%). All R^2^ for samples with TS < 8% at different RTs in Figure 2b were significantly improved compared to that in Figure 2a and the R^2^ was improved as RT increased. However, for TS > 8%, the R^2^s of NMR prediction and lab measurements in Table 2 were less than 0.23 indicating poor prediction of TN by the NMR. The TN predicted by the NMR was underestimated for low TS samples, whereas it was significantly overestimated for manure samples with high TS according to the slopes in Figure 2b. Several previous studies investigated the regression relationship between TS and TN in animal manure [22,23] indicating the performance of NMR for TN prediction might be affected by the sample TS. Another possible explanation of the effect of TS levels on the TN measured by the NMR was that ^17^O was used to determine the organic N fraction in TN and the NMR intensity of ^17^O was observed to be directly correlated to the dry matter [11]. The bias of linear regression and R^2^ of 0.5 for samples with TS > 8% indicated the assumed correlation between ^17^O NMR intensity and dry matter was not significant and thus resulted in the non-linearity of organic N and the ^17^O NMR signal. The concentration of TN in the manure sample was estimated as the sum of NH_4_-N and organic N concentrations. Therefore, a poor correlation and bias was observed for the linear regression of TN for samples with TS > 8% (Table 2). This result may be attributed to the inaccurate prediction of NMR for higher TS manure samples.

#### 3.2.3. Ammoniacal Nitrogen

Figure 3 shows the NH_4_-N measurements in the laboratory and using NMR at different RTs. The NMR predictions for NH_4_-N were highly correlated to the laboratorial analysis with R^2^ greater than 0.94 for all RTs (Table 2) showing good performance of NMR in predicting NH_4_-N in manure (Figure 3a). The intercepts of the linear regressions of NMR prediction versus lab measurement were greater than 1.0 which indicated the NH_4_-N predicted by NMR was higher than that measured in the lab generally.

The residuals of NMR predicted and measured distributed homogeneously within ±200 mg L^−1^ except for three samples with TS between 10% and 20% (Appendix A). When linear regression was adjusted and modeled based on the TS of samples greater and less than 8% as discussed above, the R^2^ values for TS < 8% in Figure 3b were improved compared to that in Figure 3a and were greater than 0.98 for all RT scenarios (Table 2). Comparing R^2^s of two TS levels for NH_4_-N in Table 2, the R^2^s for TS < 8% were significantly higher than that for TS > 8% at four RTs which indicated the NMR prediction of NH_4_-N was more accurate for samples with relatively low TS.

The TS levels had less impact on NH_4_-N than on TN measurements since the NH_4_-N was determined by a selective measurement of nitrogen dissolved in animal manure based on the ^14^N NMR. However, the NMR predictions for manure samples with TS < 8% were better than those with TS > 8% on TS, TN, and NH_4_-N, respectively, and this illustrated that the accuracy and precision of the NMR analyzer for predicting TS, TN, and NH_4_-N were affected by the TS level of samples and the NMR reading may be robust within a limited range of TS. This may be explained by the calibration of the NMR system. According to the data reported by Sørensen et al. [11], samples of 16 animal slurries collected from cattle, pigs, mink, and biogas plants were used for the calibration of the NMR analyzer and the means (ranges) for NH_4_-N, TN, TP, and TS of samples were 1991 (300–4600) mg L^−1^, 3184 (400–6000) mg L^−1^, 697 (100–2400) mg L^−1^, and 4.0 (0.2–9.4)%, respectively. The TS of manure samples used for calibrating the analyzer was relatively lower than the dairy manure used in this study (TS: 1.4–19.8%), and the manure compositions were more concentrated with NH_4_-N, TN, and TP than the manure samples selected in our research (NH_4_-N: average 1301 with range of 751–4253 mg L^−1^; TN: average 2814 with range of 1114–5824 mg L^−1^; TP: average 506 with range of 155–1097 mg L^−1^). The NMR device used in this study was pre-calibrated by the manufacturer for each parameter, therefore, the calibration of the NMR system is accurate only for a restricted range of measurements. The manure samples predicted in this study with relative high TS may be beyond the calibration curve of the system and result in bias and system errors in applications.

#### 3.2.4. Total Phosphorus

The TPs predicted by the NMR showed good agreement with the laboratory measurements and the R^2^s of the linear regressions were 0.89, 0.92, 0.91, 0.88 for RT 30, 45, 60, and 90 min, respectively (Table 2). The correlation between the NMR prediction and lab measurement was not enhanced as the RT increased, and the NMR had the best performance on predicting TP at RT 45 min. The intercepts of the linear models were less than 1.0 at all RTs showing the evidence of underestimation of TP by the NMR analyzer.

The residual plot showed a good homogeneity of the linear regression between NMR and lab measurements (Appendix A) with only one sample having large variations. Compared to the models including all samples, the R^2^s for TS < 8% and TS > 8% at all RTs (Figure 4b) were smaller than those in overall prediction (Figure 4a) with the exception of TS > 8% at RT 30 min (Table 2) which indicated that grouping the data based on TS levels of TS > 8% and TS < 8% did not improve the performance of the linear relationship between NMR prediction and laboratorial analysis and the effect of TS on TP prediction of NMR was not significant. Large solid contents may result in nonhomogeneous distribution of nutrients in manure and the uncertainties due to subsampling challenging may be significant in TP analyzing. However, ^31^P NMR with abundant spin-1/2 ^31^P isotope was suitable for detecting both dissolved and solid P components in manure [11]. This may be regarded as an advantage of an NMR analyzer over traditional chemical analysis for TP in manure.

## 4. Discussion

### 4.1. The Effect of Run Time (RT) on NMR Prediction for Manure Nutrients

The TS was measured at an RT of 10s for all samples and the overall AbsDiff mean of TS based on 19 samples (S11 was excluded as an outlier) was 6.1 ± 3.66% which indicated that TS estimated by NMR was 6.1% greater than the lab TS on average (Table 3). The AbsDiff means for TN were 896.4 ± 407.1, 839.0 ± 373.2, 814.6 ± 327.7, and 856.0 ± 382.1 mg L^−1^, at RT 15, 30, 45, and 60 min, respectively (Table 3). As RT increased from 15 to 45 min, the AbsDiff of means and SDs for TN decreased indicating that the NMR measured accuracy and precision were both improved, however, as RT further increased to 60 min, the AbsDiff of mean and SD for TN increased, resulting in less accuracy and precision of NMR measurement. The accuracy of NH_4_-N prediction continuously improved as the RT of NMR increased from 15 to 60 min with the means of AbsDiff dropping from 359.9 to 306.6 mg L^−1^. However, the precision of the prediction improved as RT increased from 15 to 45 min, and the SD of RT 60 min (195.2 mg L^−1^) was greater than that of RT 45 min (163.6 mg L^−1^) indicating a less precise prediction of NH_4_-N at RT 60 min compared to RT 45 min. The TP prediction of NMR was most accurate when the analysis was run at RT 90 min with the smallest AbsDiff mean of 68.4 mg L^−1^ (Table 3). However, the accuracy of the TP prediction fluctuated up and down rather than a continuous improvement as RT increased from 30 to 90 min. The TP prediction at RT 30 min had the best precision among the four RTs with an SD of 35.0 mg L^−1^.

Although the NMR analysis is considered to have lower sensitivity, the signal scales linearly on repeating the experiments over time by adding more data whereas the noise scales with the square root of the number of experiments which means the precision of the NMR analysis can be increased by running the experiments for a longer RT. However, the analysis of AbsDiff between NMR and lab measurements did not show consistent results of the RT effect on the precision of NMR predictions for TS, TN, NH_4_-N, and TP (Table 3). The results showed running experiments for a longer time had a positive effect on increasing the precision of NMR for TN and NH_4_-N, but when the RT was further elongated the prediction did not improve. A relationship between RT and NMR precision especially for TP was not observed. This may be due to the stronger effects of solids precipitation when measuring at a longer time or non-homogeneous distribution of particles in manure [19,24].

### 4.2. Repeatability and Reproducibility of NMR Manure Prediction

In Table 3, the R_p_ for AbsDiff of TS was 60% indicating a large variation among samples which meant the NMR prediction for TS was not consistent among different manure samples and the reproducibility of NMR for TS measurement was poor. The precision of intra-sample represented by R_p_ was 18.6% for TS which performed much better than the inter-sample precision indicating a good repeatability when the analysis was replicated for an individual manure sample.

The R_p_ and R_d_ for TN had a similar trend among the four RT groups. When TN was measured at 45 min RT, R_d_, and R_p_ had the best performances which were 40.2 and 15.1%, respectively. The intra-sample precision (R_p_) of NH_4_-N was improved from 48.6% at RT 15 min to 31.0% for RT 60 min. The overall R_d_s for all RT groups were relatively high of over 50%, which indicated the inter-sample variations for NH_4_-N measurements were large. The repeatability and reproducibility of the TN analysis were better than those of NH_4_-N which indicated the NMR analysis of TN was more precise and robust when compared to NH_4_-N prediction. The intra-sample CVs were greater than 56.6% for TP at different RTs which were relatively higher in comparison to the other parameters showing large variations and poor precisions of NMR in TP prediction. The overall R_p_ was significantly large which indicated the extreme variations among replicated readings of TP for individual samples and the NMR prediction of TP was not repeatable or robust. Moreover, the R_p_ for TP was even greater than the Rd which showed the poor precision of NMR for predicting TP. This was because the intra-sample CV was generally lower than the inter-sample CV due to the higher variations among different samples such as the manure characteristics. Compared to other manure components, the intra-sample variations of TP were relatively large at all RTs and this may indicate that the precision of TP predicted by NMR was more difficult to obtain. Similar results were found in laboratorial analysis of TP. Sanford et al. [2] conducted the analysis of accuracy and variability of certified manure laboratories and reported the accuracy and precision, evaluated using median absolute deviation (MAD) to median ratio of P, was significantly higher than other parameters. Sørensen et al. [11] observed large deviations in individual laboratory results for TP in animal slurries. These implied the lab measurements of TP used in this study may not be accurate and precise, and therefore, using TP measurements in the lab as the reference may result in bias and error.

### 4.3. Comparison with Existing Literature

As a potential alternative to existing manure nutrient analysis methods, it is necessary to evaluate the performance of NMR prediction based on the findings of other studies. Jensen et al. [25] analyzed manure slurries of different species using both low-field NMR and standard laboratory measurements to calculate cross method error. The SDs of NMR measurements for NH_4_-N and TN were 110 mg L^−1^ and 147 mg L^−1^ with 5 min RT, respectively which were comparable to the SD of 128.00 mg L^−1^ and 116.75 mg L^−1^ with 15 min RT for NH_4_-N and TN in this study (Table 3), respectively. The NMR SDs were reduced as the RT increased up to 45 min of NH_4_-N and TN measurements. For TP, the SD was 74 mg L^−1^ with an RT of 10 min compared to an SD of 23.57 mg L^−1^ with 30 min RT in the present study (Table 3). The NMR SDs in this study of TP were significant smaller than that reported by Jensen, however, the R_p_ indicated large variations among replicates of TP as discussed. Jensen et al. [25] investigated the instruments by operating 20 NMR devices in parallel on all samples and estimated the overall SD that contained contributions from both samples and measurements per instrument. This SD may be roughly compared to the experiment conducted by Sanford et al. [2] which was an external study to assess the effectiveness of the MAP program based on the evaluation of eight MAP certified laboratories. The average SDs of the laboratory measurements were 211 mg L^−1^, 371 mg L^−1^, and 239 mg L^−1^ for NH_4_-N, TN, and TP, respectively and the overall SDs of NMRs were 139 mg L^−1^ for NH_4_-N, 170 mg L^−1^ for TN, and 124 mg L^−1^ for TP which indicated the NMR had the potential to provide measurements of N and P in animal manure with the same levels of accuracy as the laboratory.

### 4.4. Limitations and Uncertainties

Compared to traditional chemical analysis, the NMR requires less sample preparation procedures and can measure several manure components simultaneously within a relatively short time. However, the results of this study indicate several limitations and uncertainties of this NMR sensor for manure analysis that need to be addressed before on-farm application. The nutrient contents of animal manure are highly variable, and the sample size of this study is relatively small. This limitation may affect the reliability of the results and lead to bias. Moreover, NMR sensing is a chemical measurement method that does not directly measure the suspended solid particles in manure. The TS content is indirectly estimated based on the assumed correlation between ^17^O NMR intensity and water content. Thus, the predicted TS content may result in a high relative error. This error is further transferred to the estimation of organic N which is calculated based on the assumption that organic N correlates with TS content. The weakness of this measurement approach may make the errors in TS and TN predictions relatively high.

Further work including evaluation of the NMR sensor with a larger sample size and other animal slurries, measuring TS with a direct method, improving the accuracy and precision of TP prediction, and comparing NMR measurements with multiple laboratories and other technologies (e.g., NIRS) are crucial to evaluate the application of NMR for manure sensing.

In summary, the low-field NMR was able to predict the compositions of TS, NH_4_-N, TN, and TP in dairy manure. The predictions of TS, NH_4_-N, and TN were in good agreement with the lab measurements for samples with TS less than 8%. The NMR predictions for TS, NH_4_-N, and TN were not well correlated to the lab measurements for samples with TS greater than 8%. The TP predicted by NMR was not affected by TS levels and the overall predictions showed good correlation with the lab results. The accuracy and precision of NMR prediction were improved by increasing the RT for NH_4_-N analysis, however, no consistent relationship between the accuracy and precision of the NMR analyzer and the RT was observed for all manure parameters. The intra-sample precisions (R_p_) were better than the inter-sample precisions (R_d_) due to the extra variations introduced from the differences among manure samples for TS, TN, and NH_4_-N. The intra-sample variations of TP were higher than those of TS, TN, and NH_4_-N and even higher than the inter-sample variations which illustrated the TP prediction of NMR was not precise and robust with poor repeatability and reproducibility. The results in this study illustrated several issues related to the system accuracy and precision of the NMR analyzer for predicting manure nutrients that require further study.

## 5. Conclusions

To access the accuracy and precision of an innovative low-field NMR analyzer, twenty liquid and slurry dairy manures were collected and analyzed using the NMR analyzer for TS, TN, NH_4_-N, and TP measurements in this study. Data were compared to the results from a certified laboratory. The predictions of TS, TN, and NH_4_-N using the NMR were highly correlated to the lab measurements for samples with TS < 8%. The R^2^ values for the linear regressions were 0.86 for TS and greater than 0.94 and 0.98 for TN and NH_4_-N, respectively. The TP predicted by NMR agreed with the lab analysis for all samples with R^2^ greater than 0.88. The prediction for NH_4_-N was most accurate at RT 60 min, and RT 45 min had the best precision of measurement. TN was predicted with the highest accuracy and precision at RT 45 min. TP measured at RT 90 min was most accurate, but the precision was better at a 30 min RT. TP measured by NMR had significantly large intra-sample variations. The study demonstrates the low-field NMR sensor is easy to operate without complicated sample preparations and the time required for analyzing is relatively short compared to conventional laboratory analysis. The performance indicates there is a potential for using the low-field NMR instrument as an alternative for manure nutrient analysis, however, issues such as using a larger sample size, and improving the methodology of TS measurement need to be addressed in future work.

## Figures and Tables

**Figure 1 sensors-22-02438-f001:**
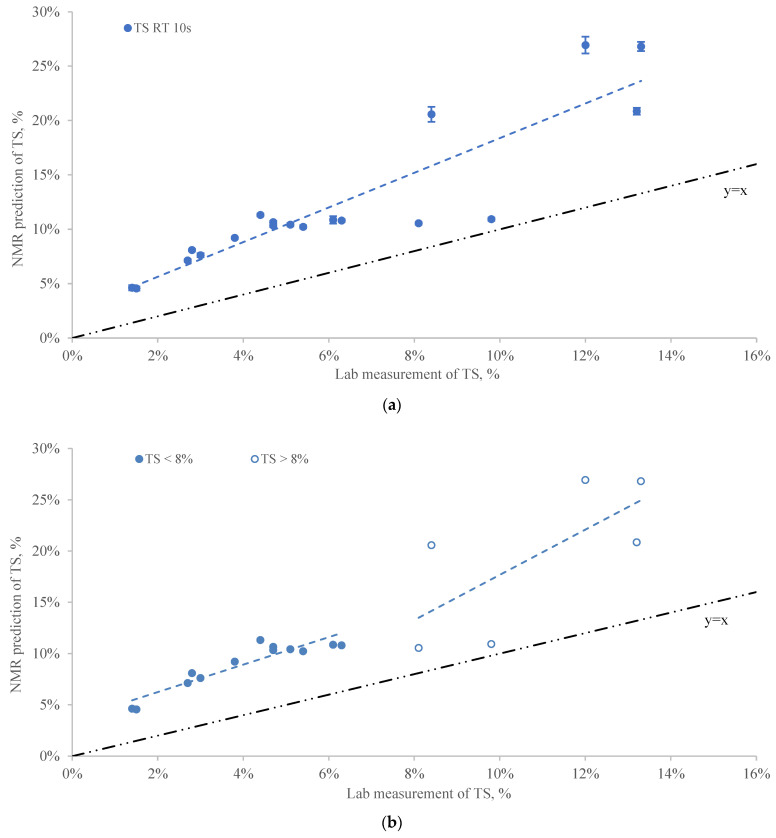
(**a**) The linear regression of NMR prediction vs. lab measurement for TS at 10 s RT (Error bars represent the standard error of 20 replicates). (**b**) The adjusted linear regression of NMR prediction vs. lab measurement for TS based on different TS groups at 10 s RT.

**Figure 2 sensors-22-02438-f002:**
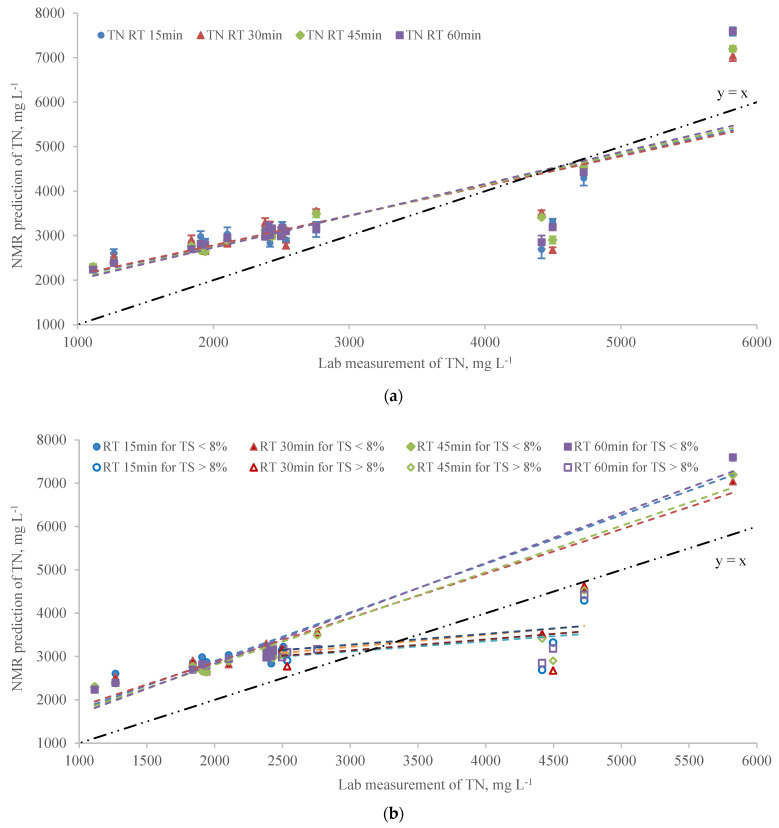
(**a**) The linear regression of NMR prediction vs. lab measurement for TN at 15 min, 30 min, 45 min, and 60 min RTs (Error bars represent the standard error of 5 replicates). (**b**) The adjusted linear regression of NMR prediction vs. lab measurement for TN based on different TS groups at 15 min, 30 min, 45 min, and 60 min RTs.

**Figure 3 sensors-22-02438-f003:**
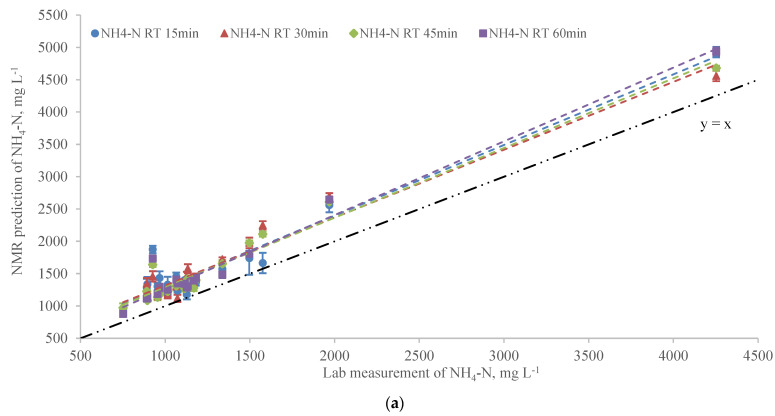
(**a**) The linear regression of NMR prediction vs. lab measurement for NH_4_-N at 15 min, 30 min, 45 min, and 60 min RTs (Error bars represent the standard error of 5 replicates). (**b**) The adjusted linear regression of NMR prediction vs. lab measurement for NH_4_-N based on different TS groups at 15 min, 30 min, 45 min, and 60 min RTs.

**Figure 4 sensors-22-02438-f004:**
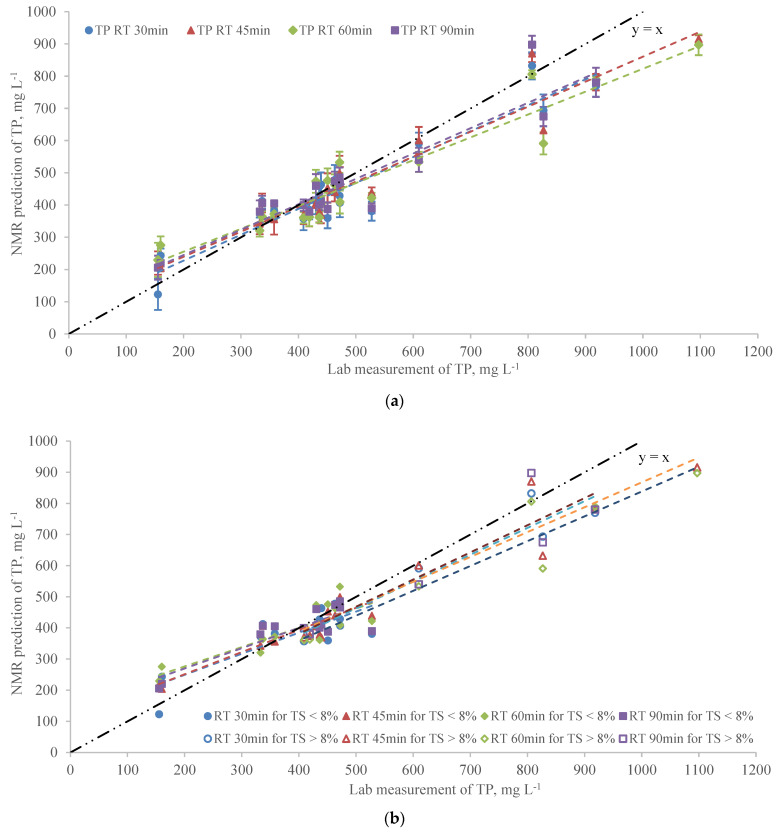
(**a**) The linear regression of NMR prediction vs. lab measurement for TP at 30 min, 45 min, 60 min, and 90 min RTs (Error bars represent the standard error of 5 replicates). (**b**) The adjusted linear regression of NMR prediction vs. lab measurement for TP based on different TS groups at 30 min, 45 min, 60 min, and 90 min RTs.

**Table 1 sensors-22-02438-t001:** Laboratory analysis of manure compositions.

Sample ID	NH_4_-N	TN	TP	TS
1	1072	2534	409	9.8%
2	1128	2497	419	8.1%
3	1128	2417	451	5.4%
4	1116	2385	436	5.1%
5	1134	2381	430	3.8%
6	1168	2431	439	4.7%
7	1181	2407	472	4.4%
8	955	1940	358	3.0%
9	892	1838	337	2.8%
10	967	1906	333	2.7%
11	1497	4332	807	19.8%
12	4253	5824	528	6.1%
13	1576	4416	918	12.0%
14	1015	2103	472	4.7%
15	751	1114	160	1.4%
16	895	1268	155	1.5%
17	1337	2757	610	8.4%
18	1065	2507	463	6.3%
19	1969	4726	827	13.2%
20	926	4497	1097	13.3%

NH_4_-N = concentration of ammoniacal nitrogen (mg L^−1^); TN = concentration of total nitrogen (mg L^−1^); TP = concentration of total phosphorus (mg L^−1^); TS = total solid content (%).

**Table 2 sensors-22-02438-t002:** Linear fitness of NMR prediction vs. lab measurements for manure nutrients based on overall samples and the adjustments of total solid groups.

Parameter	RT (min)	Overall		TS < 8%		TS > 8%	
		Linear	R^2^	Linear	R^2^	Linear	R^2^
TS (%)	10 s	y = 1.593x + 0.025	0.80	y = 1.34x + 0.04	0.86	y = 2.20x − 0.04	0.50
TN (mg L^−1^)	15	y = 0.677x + 1422	0.56	y = 1.12x + 649	0.94	y = 0.235x + 2412	0.21
	30	y = 0.667x + 1451	0.63	y = 1.02x + 819	0.96	y = 0.280x + 2385	0.18
	45	y = 0.700x + 1340	0.66	y = 1.07x + 668	0.96	y = 0.250x + 2524	0.21
	60	y = 0.717x + 1296	0.61	y = 1.16x + 518	0.96	y = 0.253x + 2382	0.23
NH_4_-N (mg L^−1^)	15	y = 1.09x + 215	0.94	y = 1.11x + 181	0.98	y = 0.859x + 554	0.51
	30	y = 1.05x + 269	0.96	y = 1.01x + 259	0.99	y = 1.45x − 170	0.90
	45	y = 1.08x + 211	0.97	y = 1.06x + 155	1.00	y = 1.16x + 233	0.84
	60	y = 1.14x + 125	0.97	y = 1.14x + 85	1.00	y = 1.11x + 255	0.70
TP (mg L^−1^)	30	y = 0.803x + 67	0.89	y = 0.677x + 114	0.68	y = 0.862x + 32	0.91
	45	y = 0.776x + 84	0.92	y = 0.711x + 109	0.84	y = 0.799x + 69	0.87
	60	y = 0.710x + 113	0.91	y = 0.62x + 153	0.72	y = 0.798x + 41	0.90
	90	y = 0.788x + 87	0.88	y = 0.652x + 139	0.76	y = 0.873x + 32	0.84

RT = run time; TS = total solid content; NH_4_-N = concentration of ammoniacal nitrogen; TN = concentration of total nitrogen; TP = concentration of total phosphorus.

**Table 3 sensors-22-02438-t003:** Statistical analysis of AbsDiff between NMR and lab measurements of manure composition at different run times.

Parameter	RT	N	Mean ± SD (AbsDiff)	SD (NMR Measurements)	R_d_ (%)	R_p_ (%)
TS (%)	10 s	19	6.1 ± 3.66	0.88	60.0	18.6
NH_4_-N (mg L^−1^)	15 min	20	359.9 ± 204.47	128.00	56.8	48.6
	30 min	20	337 ± 169.02	46.10	50.2	42.3
	45 min	20	312.4 ± 163.62	28.21	52.4	33.4
	60 min	19	306.6 ± 195.16	42.30	63.7	31.0
TN (mg L^−1^)	15 min	19	896.4 ± 407.08	116.75	45.4	24.9
	30 min	19	839 ± 373.15	54.48	44.5	20.6
	45 min	19	814.6 ± 327.67	40.66	40.2	15.1
	60 min	19	856 ± 382.12	76.85	44.6	16.3
TP (mg L^−1^)	30 min	19	78.3 ± 35.01	23.57	44.7	73.9
	45 min	20	81.6 ± 43.04	27.87	52.8	70.2
	60 min	20	85.2 ± 54.57	24.31	64.0	56.6
	90 min	19	68.4 ± 37.75	23.01	55.2	73.4

RT = run time; N = sample size; SD = standard deviation; TS = total solid content; NH_4_-N = concentration of ammoniacal nitrogen; TN = concentration of total nitrogen; TP = concentration of total phosphorus; R_d_ = reproducibility; R_p_ = repeatability.

## Data Availability

Not applicable.

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
