# Peer review of "Evaluating the Feasibility of a Low-Field Nuclear Magnetic Resonance (NMR) Sensor for Manure Nutrient Prediction"

_sensors, 2022, doi:10.3390/s22072438_

Round 1

Reviewer 1 Report

The authors presented a study to compare the performance of a recently NMR system presented to the market to measure various nutrients in manures. The study tends to be more a calibration of a new system rather than a comparison of the system performance to the existing methods. The authors need to take into account the following comments to enhance the manuscript quality.      

ABSTRACT:

Line 17:  The authors need to state the values as the word “greater than” is confusing and not accurate.

INTRODUCTION

The authors need to state other non-invasive methods used to measure manure nutrients other than NMR. How about optical methods? FTIR, Raman, etc.   

Lines 63-66: The authors need to state more details about the way the system works and why repeatability is not consistent.

Lines 69-72: The authors need to state the results obtained by the authors and why this study is not considered enough to evaluate the system performance.

Materials and Methods

In general, 20 samples is not enough to judge the performance of a measurement system.  The authors need to justify why more samples were not used.   

Line 87: add respectively.

Line 101:  State what PFA and PTFE stand for.

Lines 105-106: Does that mean the test for each sample was repeated 5 times? If so, this sentence needs to be rewritten.   

Line 130-132: The authors need to provide a reference for these actions.

Lines 132-134: It will be more appropriate to place all of discarded samples for each parameter in a table.

Result

The authors need to justify why the TS estimation was conducted in 2 regions (<8% and >8%).

Discussion

The authors didn’t include any comparison to other studies using different systems which is a true drawback of the study. This is a calibration of the system type of study.

Reviewer 2 Report

In general the topic of the paper, manure analysis for fertilizing, is very important from many points of view. However, the contributions of the paper are modest. Please find more detailed comments below.

Title: is quite general. Could be focused by telling what 'manure analysis' mean here.

Abstract: tells quite nicely about the motivation, methods, and results. As already said above, the contribution and conclusions thereon are modest.

Keywords: low-field NMR is missing.

Introduction:

Chapter 1: OK

Chapter 2: Could tell what chemical analysis are needed and how they can be done using NMR. Introduces a new NMR instrument, but does not tell much about it. NMR is a 'workhorse' of chemical labs. The authors could tell how this high precision technology is used to manure analysis, what are the main challenges, strengths, and weaknesses. To do measurements successfully means that the measurement principles should be known. Here is difficult to see such information at all. Even the device makers webpages does not reveal much. The device just a real black box, in side of which is an instrument that able to do some NMR-spectroscopy. The authors should tell what are the principles of device and how those influence the expected results.

On lines 59-61 the authors tell the main parts of the device. However, they do not tell anything about the capabilities of the device. The authors should give more detailed information on the device.

On lines 66-73 the authors refer to one paper that has tested the device. After having read the paper, it seems that the authors has not got much more test results of practical use.

Materials and methods

The approach of the paper is that of statistics. It would be nice if the authors would use more samples to get more reliable statistics.

Laboratory and NMR analysis

The reference measurements were done in a local lab. What is the precision of the results of that lab. The authors should tell more about the NMR device and its claimed accuracy. Could tell the magnet strength (Teslas etc).

The device has been pre-calibrated by the manufacturer. How that was done? Namely the NMR devices typically give very complex and precise chemical information. How that information was collected and transformed into the measurement results? That information helps to understand what can be measured and at what precision. Manure is quite far from simple chemical analysis stuff.

Data analysis:

5 replicates sound quite low for basic statistics.

What was the error tolerance of the lab measurements and for NMR?

Do the NMR maker give any such information?

What was the expected reason for the outliers?

Results:

Table 1 gives results with 5 digits. Are the measurements really so precise? Please give realistic estimates of the error tolerances within which the measurements are most probably.

TS is given in Table 1 with only 2 digits?!  Also for it the authors could estimate the error tolerances.

BTW, what solids mean here chemically? NMR is good in chemistry but not that good in solid states...

Comparison:

Figure 1a: It seems that there is a bias (at zero non-zero result).

What is the reason for that?

Figure 1b: Residuals: just the Figure 1a rotated so that the regression line is x-axis. What is the information that can be seen here, that is not evident in Figure 1a?

Figure 1c: Seems to be an attempt to explain some nonlinearity. Why not simply do non-linear regression e.g. by a polynomial?

Table 2. What is the reason for the non-linearity? Is it so that the solids do interfere in measurements (the sample should be a solution in NMR)?

Figure 2a: Also here a clear bias. What would be the reault if there is no N?

(measure just pure water).  What N the device is actually measuring?

Figure 2b: like Figure 1b: what is the new information w.r.t. a figure?

Figure 2c: why not polynomial regression?

Ammoniacal nitrogen:

Figure 3a looks very nice (not much bias here). Is the conclusion that NMR is really good in doing fine chemical analysis? (and not so good with cruder analysis). TS has less effect here. Good. However, the authors claim that this might be due to the calibration of the device: the authors must be right.

However, there is no information how the calibration was actually done at the NMR spectrum level. Obviously the (fine) peaks of the spectra should be located and accumulated to give the (crude) measurement results.

More information on the principles of operation of the device is needed.

Figure 3b: line Figure 2b and 1b: not much new information.

Figure 3c: OK. NMR really finds ammonia. The referee is not surprised.

Figure 4a: OK (bias very small).

Figure 4b: any new information here? BTW, should this be phosphorus not ammonia?

Figure 4c: phosphorus?

Discussion:

Is the problem (weakness of the device) measuring TS with NMR?

Table 3: OK

How about the sedimentation when measuring long times? Do the solids float or sink or both? (sample dependent?)

Lines 356-357: the authors should ask the lab about their measurement errors.

Lines 375-384: There are two or more calibration procedures:

First the 'fine spectra' should be mined for the 'crude' chemistry results.

That is something the maker is supposed to do.

Then the user should make calibrations with known manure samples.

Obviously this needs some NMR and chemistry knowledge from the user.

---------

Summa summarum: for successful measuring the device and its principle of operation (with the sample) and data processing should be known.

Reviewer 3 Report

This paper analyses with a low field NMR analyzer twenty dairy manures via measurements of TS, TN, NH4-N, and TP. The collected TS, TN, and NH4-N data compared to the results of a certified laboratory showed a strong correlation with the laboratory measurements. Authors have noted and pointed out the difficulty of measuring PT compared to the other parameters, as the latter showed significantly large intra-sample variations. The paper’s subject is interesting. The paper presentation is well. The result and method are presented clearly. The research work presented by the authors remains original. Overall, the results are interesting and deserve to be published in the journal sensors in the present form.

Round 2

Reviewer 1 Report

The authors clearly replied to the previous comments. However, I have one suggestion to get the manuscript quality to the best level possible.  

The authors claim that the study is to “examine if the reported accuracy and precision of the manufactured calibration of the NMR are robust and appropriate for manure samples”. They also claim this sample size represents the Wisconsin State well. The authors need to state the geographical distribution of the samples’ sources. Additionally, the authors need to

  • Change the title to add the word feasibility which refers to the possibility of the idea to be correct in general and NOT to be applied in a robust way.
  • Add a couple of sentences to the discussion section that clearly illustrates the drawback of using such a small sample size.
  • Compare with a previous study or studies that sued a similar small sample size. This needs to be added to the Materials and Method section (Manure sample collection).
  • Add to the conclusion section a sentence to state the need to use a larger sample size for future work.

Reviewer 2 Report

Thank you for the revisions.

They are mostly quite OK.

However, one key point is the problem of TS / NMR.

How it is actually measured: indirectly by measuring water.

Obviously the amount of water is very high i.e. approaching 100%.

It means that the TS = 100 - (W ~ nearly 100)  making the relative error in TS high. This is the weakness of this measurement approach of both theoretical and practical nature: using this approach the error in TS will be always high.

This is because NMR does not (at least in this (cheap) device) have any direct way of sensing 'solid'. NMR is a chemical measurement method for which 'solid' is not well defined at all.

My proposal is that the authors clearly state this fact or present a convincing statement how the 'TS' would be well defined for this device. An alternative is to do TS measurements by some other method than NMR.
